# Immunogenicity and Immunoprotection of PCV2 Virus-like Particles Incorporating Dominant T and B Cell Antigenic Epitopes Paired with CD154 Molecules in Piglets and Mice

**DOI:** 10.3390/ijms232214126

**Published:** 2022-11-16

**Authors:** Keke Wu, Wenshuo Hu, Bolun Zhou, Bingke Li, Xiaowen Li, Quanhui Yan, Wenxian Chen, Yuwan Li, Hongxing Ding, Mingqiu Zhao, Shuangqi Fan, Lin Yi, Jinding Chen

**Affiliations:** 1College of Veterinary Medicine, South China Agricultural University, Guangzhou 510642, China; 2Key Laboratory of Zoonosis Prevention and Control of Guangdong Province, Guangzhou 510642, China; 3Guangdong Laboratory for Lingnan Modern Agriculture, Guangzhou 510642, China

**Keywords:** porcine circovirus type 2 (PCV2), capsid protein, virus-like particles (VLPs), T and B cell antigenic epitopes, IFN-γ, IL-4

## Abstract

Porcine circovirus type 2 (PCV2) is capable of causing porcine circovirus-associated disease (PCVAD) and is one of the major threats to the global pig industry. The nucleocapsid protein Cap encoded by the PCV2 ORF2 gene is an ideal antigen for the development of PCV2 subunit vaccines, and its N-terminal nuclear localization sequence (NLS) structural domain is essential for the formation of self-assembling VLPs. In the present study, we systematically expressed and characterized full-length PCV2 Cap proteins fused to dominant T and B cell antigenic epitopes and porcine-derived CD154 molecules using baculovirus and found that the Cap proteins fusing epitopes were still capable of forming virus-like particles (VLPs). Both piglet and mice experiments showed that the Cap proteins fusing epitopes or paired with the molecular adjuvant CD154 were able to induce higher levels of humoral and cellular responses, particularly the secretion of PCV2-specific IFN-γ and IL-4. In addition, vaccination significantly reduced clinical signs and the viral load of PCV2 in the blood and tissues of challenged piglets. The results of the study provide new ideas for the development of a more efficient, safe and broad-spectrum next-generation PCV2 subunit vaccine.

## 1. Introduction

PCV2 is the smallest animal virus ever discovered, with free viral particles of approximately 17 nm in diameter and a symmetrical, icosahedral structure without a capsule [1]. The PCV2 genome is a single-stranded, negative-stranded, circular DNA containing two major open reading frames (ORFs)—ORF1 encoding two replication-associated proteins (Rep and Rep’) and ORF2 encoding the viral capsid protein (Cap) [2]. PCV2 is the main pathogen causing post-weaning multiple system failure syndromes (PMWS) and porcine circovirus-associated disease (PCVAD). PCV2 is arguably an important pathogen affecting the global pig industry and the main pathogen of PCVAD. Currently, PCV2 infections are on the rise in countries around the world, and PCVAD caused by PCV2 leads to immunosuppression, which affects production performance and causes huge economic losses to global pig production.

The control of PCVAD in China is still mainly by vaccination. Cap is the main immunogenic protein that can be used to induce PCV2-neutralizing antibodies and is an ideal antigen for the development of PCV2 subunit vaccines [3,4]. The PCV2 cyclic genome is wrapped in 60 Cap protein subunits that are arranged in 12 pentameric aggregation units to form homopolymeric icosahedral viral particles [5]. The NLS structural domain at the N-terminal end of the Cap protein is essential for the formation of its self-assembling virus-like particles (VLP) [6]. Cap protein-based virus-like particle vaccines have been developed using expression systems, such as *E. coli* and baculovirus [7,8,9,10]. VLP vaccines have a similar spatial three-dimensional structure and composition to natural viral particles and are able to stimulate the immune system through a pathway similar to that of viral infection and efficiently induce an immune protective response in the body, thus offering great advantages as vaccine candidates for prevention and control.

The onset and severity of PCVAD are associated with a lack of or reduced levels of PCV2-neutralizing antibodies, suggesting that neutralizing antibodies play a crucial role in the prevention of PCVAD and that the ORF2-encoded Cap was chosen precisely because it contains the major neutralizing epitope [11,12,13]. It has been shown that protective antigens have synergistic effects with multi-epitope fusions, such as multiple T and B cells, potentially inducing a more active humoral and cellular immune response [14,15,16,17]. Although the use of CD154 (CD40L) as a molecular adjuvant for vaccines is at an exploratory stage [18,19,20], CD154 has been shown to significantly improve the immune efficacy of vaccines in studies of Human Respiratory Syncytial Virus, Bovine Herpesvirus and Duck Hepatitis B Virus [21,22,23,24,25]. Therefore, in our study, we propose to develop a novel genetically engineered subunit vaccine for PCV2 by fusing the dominant T and B cell antigenic epitopes of PCV2 and introducing the molecular adjuvant CD154 protein in two different strategies. 

Compared to the prokaryotic expression systems such as that in *E. coli*, the currently established baculovirus expression system has a more complete protein structure modification function and is one of the best options for expressing vaccines against VLPs. In this study, we used a baculovirus expression system to express a full-length PCV2 Cap protein fused to a dominant T and B cell antigenic epitope and a molecular adjuvant: porcine-derived CD154. By evaluating immunization in mice and piglets, we expect to provide new ideas for the development of a more efficient, safe, cost-effective and broad-spectrum PCV2 subunit vaccine.

## 2. Results

### 2.1. Screening of Antigenic Epitopes and Construction of Shuttle Plasmids

PCV2 ORF2 encodes the Cap protein, the main structural protein of the virus, which is the main immunoprotective antigen of the virus, induces a specific immune response in the animal organism and contains virus-specific antigenic epitopes. Two dominant T-cell epitopes on the PCV2 Rep protein and four dominant B-cell antigen epitopes on the Cap protein identified in the laboratory were used as epitopes for this study [26], and the secondary structures of the screened dominant antigen epitope peptides were analyzed and predicted using DNAStar Protean software (Figure 1B). Considering that the nuclear localization sequence (NLS) at the N-terminal end of the Cap protein is essential for the formation of VLP and that the immunogenicity of VLP formed by Cap protein alone is poor [6], we constructed pFBD-TBCap for fusion expression of the screened dominant antigenic epitope with the full-length Cap protein (Figure 1A), with the aim of self-assembling to form VLP carrying the dominant antigenic epitope. In addition, pFBD-CD154 was constructed for the molecular adjuvant CD154, which was expected to further enhance immunogenicity. The shuttle plasmids carrying the target genes were identified by M13 F/R and were successfully transposed to baculovirus.

### 2.2. Identification of Recombinant Protein Expression in Baculovirus and Optimization of Conditions

When the recombinant baculovirus plasmid was transfected into sf9 insect cells at the logarithmic growth stage, after 72 h of transfection, it could be observed under an inverted microscope that the cells were obviously enlarged, the nuclei were swollen, the vesicles and granules were increased inside the cells, the refractive index was reduced, and in severe cases, the cells were ruptured, shed and died; while the normal cells were well attached to the wall and grew in a reticulated manner, the cytosol was translucent and clear, and the cell dish was full (Figure 2A). F3 viruses were collected and named Ac-Cap, Ac-TBCap and Ac-CD154 in that order, and virus titers were controlled by purification and concentration to 10^9^ PFU/mL.

The recombinant baculovirus described above was infected with High Five cells, the infected cells were lysed and labeled with His antibody against the recombinant protein and subjected to indirect immunofluorescence analysis, which revealed that all three antigens (Cap, TBCap and CD154 protein) could be observed as green fluorescence (Figure 2B). To further characterize the expression of the recombinant proteins, the expression of the antigens in the cells was examined by Western blot using PCV2-positive sera, anti-CD154 and anti-His-tagged mAb, respectively. The results showed that Cap, TBCap and CD154 (~28 and 44 kDa, respectively) with the expected molecular weights were efficiently expressed in High Five cells (Figure 2C). In contrast, no bands were detected in cells infected with wild baculovirus. In order to further improve the antigen expression efficiency, we optimized the expression conditions, such as infection dose and infection time of recombinant Cap and TBCap, and determined that the optimal expression condition was to harvest High Five cells expressing Cap and TBCap at 96 h after infection with an infection dose of M.O.I. = 2.0. The expression of Cap and TBCap was determined to be 349 µg/mL and 330 µg/mL, respectively, using serial concentrations of recombinant His-tagged proteins as standards, and both had a small proportion (1/4 to 2/5) of the recombinant protein secreted into the cell culture supernatant, with the majority still present in the cells.

### 2.3. Structural Modeling and Assembly of PCV2 VLP

To explore whether full-length PCV2 Caps can still assemble into VLPs after fusing a dominant antigenic epitope, we predicted the structure by modeling within PyMOL. The 3D structures of the Cap and TBCap were retrieved from the PDB database (https://www.rcsb.org/, accessed on 1 August 2022) (accession number 3JCI). Under these conditions, the crystal structure of PCV2 Cap showed an icosahedral VLP structure consisting of 60 coat protein subunits (Figure 3A). The crystal structure with TBCap as a subunit was also able to show the typical icosahedral particle folds, while different antigenic peptides were displayed in different colors on the PCV2 coat surface (Figure 3B). To further validate our predicted crystal structure features, we purified the expressed Cap and TBCap proteins by sucrose density gradient centrifugation, and the purified protein samples were subjected to transmission electron microscopy to observe the in vitro assembly of PCV2 VLP. Under transmission electron microscopy, uniform particles of PCV2 VLPs with an average diameter of approximately 17 nm were observed, together with particles of PCV2 VLPs fused with dominant T and B antigen epitopes with an average diameter of approximately 20 nm (Figure 3C).

### 2.4. Screening of Immune Adjuvants and Immune Response in Mice

To evaluate the humoral immune response induced by TBCap in mice under two different adjuvant formulations of Montanide^TM^ GEL 01 and ISA 201VG, we measured PCV2-specific antibody levels by indirect ELISA and the results were expressed as OD values at a 40-fold serum dilution. As shown in Figure 4, the anti-PCV2 antibodies in the vaccination groups prepared with the different adjuvant mix all reached high levels at 14 d after the first immunization, notably, the Cap and TBCap groups mixed with GEL 01 adjuvant had statistically significantly higher anti-PCV2 antibodies at 14 d after the first immunization and remained at high levels at 21 d and 28 d. The Cap and TBCap groups prepared with 201VG adjuvant reached the highest level of anti-PCV2 antibody at 28 d after the first immunization. It was shown that the aqueous adjuvant GEL 01 mixed vaccine group was able to induce a high level of the humoral immune response within a relatively short period after immunization and that the adjuvant 201VG (W/O/W) emulsified vaccine group was also able to induce a high level of humoral immune response after boosted immunization and that the immune efficacy lasted longer. Furthermore, the results show that the recombinant baculovirus-expressed TBCap protein has better immunogenicity than the Cap protein in the presence of the same adjuvant.

### 2.5. Quality Test of PCV2 Subunit Vaccine

#### 2.5.1. Physicochemical Property Examination

The PCV2 GEL01 subunit vaccine appears as a white, translucent, well-textured, aqueous solution. The vaccine was placed at 4 °C and observed for vaccine stability, with no obvious layering and precipitation phenomenon within 60 d, showing that the vaccine stability qualified.

The PCV2 201VG subunit vaccine appears as a milky white emulsion. A small amount of emulsion was dropped onto the surface of cold water by pipetting; the first drop quickly showed cloud-like diffusion, and the second did not diffuse after the drop, showing that the vaccine dosage form is a water-in-oil-in-water (W/O/W) biphasic oil emulsion. A total of 1 mL of the vaccine was taken into a centrifuge tube and centrifuged at 3000× *g* for 15 min, and the vaccine was stratified but not broken, indicating that the stability of the vaccine qualified.

#### 2.5.2. Security Checks

Five 1-month-old mice were vaccinated with subunit vaccines prepared in two types of adjuvants, and on 21 dpi, the mice were all healthy, with a normal mental appetite and no abnormal responses, such as swelling and inflammation, caused by the vaccine.

### 2.6. Immunogenicity and Attack Protection Experiments in Piglets

#### 2.6.1. Immunogenicity of Piglets

Piglets were immunized with subunit vaccines prepared with GEL01 adjuvant and Cap protein and recombinant TBCap protein, respectively, to evaluate antigenic immunogenicity (Figure 5A). Specific antibodies were measured by indirect ELISA and results were expressed as S/P values at a 40-fold serum dilution (Figure 5C). The results showed that both the Cap protein and recombinant TBCap protein immunized groups were able to stimulate the piglets to produce specific antibodies against PCV2 to different degrees, and the antibody levels in the recombinant TBCap protein immunized group were statistically significantly higher than those in the Cap protein immunized group 2 weeks after the second immunization. It was shown that both the PCV2-Cap protein and PCV2-TBCap expressed by the recombinant baculovirus system had good immunogenicity and were able to induce a humoral immune response in piglets.

#### 2.6.2. Temperature and Clinical Signs

Body temperatures of piglets were measured for two consecutive weeks after the challenge (as in Table 1 and Figure 5B) and no significant fever (>40 °C) was observed in any of the test groups at any time point, except for piglets 766# in the Cap group and 780# in the recombinant TBCap group, which showed a slight temperature response at 1 day post-challenge (DPC). From the clinical manifestations, the Cap group showed no obvious clinical signs except for piglets 741# which showed slight diarrhea at 8 DPC and then improved; the recombinant TBCap group showed no obvious signs except for piglets 780# which showed slight diarrhea at 6 DPC and then improved; while the PBS control group of piglets 733#, 746# and 334# showed typical clinical signs of loss of appetite, coarse coat, slow movement and depression.

#### 2.6.3. PCV2 Viremia and Viral Load in Lymph Node Tissue

PCV2 viremia was detected by conventional PCR. The results are shown in Table 1; at 14 DPC and 28 DPC, all immunized groups tested negative for PCV2 viremia in pig sera; the PBS group had 5/5 piglets positive for PCV2 at 14 DPC and 3/5 piglets positive for PCV2 at 28 DPC. At 28 DPC, each pig in the Cap and TBCap groups showed detectable PCV2 viral DNA gene copy numbers of 1.71 × 10^7^ copies/g and 1.45 × 10^6^ copies/g (group average) in their inguinal lymph nodes, both statistically significantly lower than the PBS control group (2.45 × 10^11^ copies/g) (*p* < 0.05) and the viral load present in the inguinal lymph nodes of the recombinant TBCap protein group was significantly lower than that of the Cap protein group, indicating that both immunization groups largely reduced the PCV2 load in the lymph nodes (Figure 5D).

#### 2.6.4. Histopathological Observations on Piglets after Challenge

At 28 DPC, all piglets were dissected pathologically and observed for histopathology. The lungs, lymph nodes and kidneys were essentially normal in the Cap and TBCap groups, while the lungs in the PBS group showed swelling, hemorrhage and diffuse lesions, enlarged lymph nodes and the kidneys were oedematous and pale. The sections were subjected to hematoxylin-eosin staining (HE) and histopathological examination. The results showed that thickening of the alveolar wall, widening of the interstitium and inflammatory cell infiltration were seen in the PBS group compared to the immune group, while the vaccine immune group was more normal (Figure 6A). The results suggest that the subunit vaccine prepared in this study can reduce the lesions in the lung and lymph nodes caused by PCV2 infection.

Immunohistochemical (IHC) detection was performed on piglets’ inguinal lymph nodes using PCV2 mAb with an unincubated antibody control (Figure 6B). The results showed that there were more tan-colored cells in the inguinal lymph node tissue in the PBS group, while the percentage of colored cells specific to the Cap and TBCap protein groups was significantly reduced, indicating that the Cap and recombinant TBCap protein subunit vaccines prepared in this study could largely attenuate the proliferation of PCV2 in the lymph nodes and thus provide immune protection to the immunized piglets.

### 2.7. Study on the Immunological Activity of Porcine-Derived CD154 Protein

#### 2.7.1. Immunization Strategies and Immunogenicity in Mice

We immunized mice with the prepared CD154 as a molecular adjuvant together with the antigens Cap and TBCap and performed an attack test (Figure 7A). The level of specific antibodies in mouse serum was determined by indirect ELISA and the results were expressed as OD_450_ at a 40-fold serum dilution. The results showed that after the first immunization and after the second immunization, the Cap, TBCap, CD154+Cap and CD154+TBCap groups were able to stimulate mice to produce specific antibodies against PCV2 to different degrees, except for the PBS control group, and there was no statistically significant difference between the immunized groups. After the third immunization, antibody levels continued to increase in all immunized groups, with the two groups containing TBCap having significantly higher antibody levels than the two groups containing Cap protein (*p* < 0.05), and both groups containing CD154 protein having significantly higher antibody levels than the group without CD154 protein (*p* < 0.05) (Figure 7B). The above results demonstrated that the PCV2-TBCap protein prepared in this study had better immunogenicity than the PCV2-Cap protein and was able to stimulate the mice to produce a higher level of the humoral immune response, and the porcine-derived CD154 molecule had a significant immune-enhancing effect on both Cap and TBCap antigens.

#### 2.7.2. Average Daily Weight Gain

There was no statistically significant difference in the average daily weight gain (ADWG) of the mice in each group, indicating that the immunization groups did not significantly promote the weight gain of the mice post-challenge (Figure 7C).

#### 2.7.3. Levels of Lymphocyte Proliferation

The level of lymphocyte proliferation is one of the most important indicators of cellular immune responses. To investigate whether TBCap protein and co-expression of CD154+Cap/TBCap protein can induce higher specific cellular immunity against PCV2, we examined the specific splenic lymphocyte proliferation in immunized mice using proliferating PCV2 as a stimulating antigen (Figure 8A). The results showed that lymphocyte proliferation was detected in the Cap, TBCap, CD154+Cap and CD154+TBCap groups, except for the PBS control group, and immunized mice in the TBCap and CD154+Cap/TBCap groups obtained higher levels of specific splenic lymphocyte proliferation than the Cap group (*p* < 0.05), indicating that both the TBCap protein and the TBCap group, but not the CD154+TBCap group, induced a stronger cellular immune response, probably because the multi-antigenic epitopes and CD154 molecules were not additive in stimulating higher levels of cellular immunity in the organism, but rather they acted independently to enhance cellular immunity.

#### 2.7.4. The Expression Levels of Cytokines

IFN-γ and IL-4 concentrations were measured in the secretions produced after stimulation of lymphocytes to evaluate the levels of Th1 and Th2 type immune responses in immunized mice, respectively. As shown in Figure 8B, the level of IL-4 secretion from stimulated lymphocytes was significantly higher in all immunized groups than in PBS-treated mice (*p* < 0.05), with the Cap, TBCap and CD154+Cap groups inducing a non-significant difference in IL-4 secretion levels from splenic lymphocytes (*p* > 0.05) and the CD154+TBCap group being significantly higher than the other immunized groups (*p* < 0.05), which was consistent with the results of specific antibody level detection in mice serum, indicating that the CD154+TBCap group mainly enhanced the body’s Th2-mediated humoral immune response.

As shown in Figure 8C, the levels of IFN-γ secretion from stimulated lymphocytes in the Cap, TBCap, CD154+Cap and CD154+TBCap groups of mice were all significantly higher than those in the PBS-treated mice (*p* < 0.05), with the levels of IFN-γ secretion in the CD154+Cap group being significantly higher than those in the other immunization groups (*p* < 0.05), and both the TBCap and the secretion levels of IFN-γ in both the CD154+TBCap group were significantly higher than those in the Cap group and the commercial vaccine control group (*p* < 0.05), indicating that Cap-based co-immunization with CD154 induced a bias towards Th1-type cell-mediated immune responses.

#### 2.7.5. PCV2 Viremia and Viral Load in Mice after Challenge

The qPCR method was used to quantify viremia in each test group (Figure 8D) and showed that PCV2 was detectable in the blood of all test groups at 7 DPC, with viremia reaching high levels in the PBS group, peaking at 14 DPC and decreasing to lower levels at 28 DPC. Cap, TBCap, CD154+Cap and CD154+TBCap groups maintained lower levels of the virus in the blood compared to the PBS group (*p* < 0.05). In the 14 DPC, viremia levels were significantly lower in the CD154+Cap group than in the Cap group (*p* < 0.05), but not significantly different from the other immunization groups (*p* > 0.05). At 28 DPC, the level of viremia was significantly lower in the TBCap group than in the Cap group (*p* < 0.05), but not significantly different from the other immune groups (*p* > 0.05). The results indicated that the subunit vaccine prepared in this study could not completely inhibit the occurrence of viremia but could effectively reduce its intensity, and the best effect was observed in the CD154+Cap and TBCap groups.

Viral load in different tissues of each group was determined by qPCR (Figure 8E). The results showed that PCV2 could be detected in the lung, kidney and spleen of all mice at 28 DPC, and the PCV2 gene copy number was significantly higher in the PBS group than in the immunized groups (*p* < 0.05), and there was no statistically significant difference in the viral load in the tissues of the immunized groups (*p* > 0.05). The results suggest that the subunit vaccine prepared in this study can reduce PCV2 viral load to a certain extent and is comparable to the Ingelvac CircoFLEX commercial vaccine.

## 3. Discussion

At present, PCV2 has become one of the major infectious diseases that seriously endangers the development of the world pig industry. Research on new genetically engineered vaccines that are safer, more efficient and cheaper is of great importance to prevent and control the occurrence and prevalence of PCVD.

The selection of protective antigens is crucial to ensure the immunological efficacy of novel vaccines, as Cap is the most important structural and immunogenic protein of PCV2 and the main carrier of type-specific epitopes. Several studies have shown that fusion expression of protective antigens with antigenic epitopes can better induce humoral and cellular immunity [27,28]. In our laboratory, the dominant epitopes on the Cap and Rep proteins were used to form an epitope tandem with the truncated Cap gene and cloned into the pET-32(a) prokaryotic expression vector to induce recombinant protein expression, and the specific antibody levels in immunized mice showed good immunization [29]. In this study, four B-cell antigenic epitopes at positions 61–85, 113–131, 169–180 and 192–202 aa in the Cap protein and two T-cell antigenic epitopes at positions 81–100 aa and 201–220 aa in the Rep protein were selected, taking into account the relevant reports and the conditions available in the laboratory. The full-length Cap gene, which encodes a protein closer to the natural protein structure, was selected and expressed by fusion of the epitopes with Cap through a baculovirus system. The results showed that the TBCap protein was able to self-assemble in vitro to form a VLP containing the dominant T and B cell antigenic epitopes, and that it was able to induce higher levels of antibodies and stronger cellular immune responses in subsequent animal studies.

The Bac-To-Bac baculovirus expression system was chosen for its post-translational modification function, which is suitable for producing complex recombinant proteins with a similar structure and function to natural proteins, with good safety, high expression and high biological activity of the expressed products, and can compensate for the low biological activity of the proteins expressed by the *E. coli* prokaryotic expression system and the low level of protein expression by the CHO expression system [30]. To achieve secretion and efficient expression, this study optimized the nucleotide sequence of the recombinant proteins for the codons of insect cells, introduced a bee venom signal peptide (Melittin) at the N-terminal of the target protein, and selected the more efficient High Five cells for the expression of the cell line. The optimized conditions resulted in the expression of the PCV2-Cap protein up to 349 µg/mL and the recombinant PCV2-TBCap protein up to 330 µg/mL, and the obtained recombinant proteins were all of good antigenicity. Further, the immunogenicity of the T and B cell epitope fusion Cap protein was evaluated by immunoprotection tests in mice and piglets. The results showed that both Cap and TBCap proteins stimulated the piglets to produce specific antibodies against PCV2, and the recombinant TBCap group showed significantly higher antibody levels than the Cap group after enhanced immunization and provided immune protection to the attacked piglets. It is suggested that the PCV2-TBCap protein is significantly more immunogenic than the Cap protein with the introduction of multiple antigenic epitopes and is expected to be a good candidate protein for the development of the PCV2 subunit vaccine.

Numerous studies have shown that, when co-expressed with antigens, cytokines can act as molecular adjuvants to increase the immunogenicity of vaccines and the activation of adaptive immune responses in different species, and that CD154 interacts with its receptor CD40 to play an important role in both humoral and cellular immune responses [31,32,33]. The use of CD154 as an immune adjuvant to enhance the immune efficacy of vaccines has been progressively demonstrated in human respiratory syncytial virus (RSV) gene vaccines, bovine herpesvirus gene vaccines, swine fever E2 subunit vaccines, influenza vaccines and some avian gene vaccines, but its use in PCV2 subunit vaccines has rarely been reported [21,23,24,30,34]. We further evaluated the immunoprotective effect of porcine-derived CD154 through an immunoprotection test in mice. The results showed that the immunoprotective effect of the Cap protein subunit vaccine was significantly enhanced by the introduction of the CD154 molecular adjuvant, and the mechanism may be closely related to the induction of the Th1-type cellular immune response production by CD154, stimulation of B cell activation, differentiation and Ig production by CD40-CD154 interaction and isotype interchange [35,36]. PCV2 infection induces the proliferation of IFN secretion-associated cells, including CD4^+^ and CD8^+^ T cells, and the induction of cellular immunity is necessary for the control of PCV2 infection [37]. Based on this, a CD154+ Cap subunit vaccine prepared from recombinant baculovirus is an attractive candidate for the prevention and control of PCV2 infection.

## 4. Materials and Methods

### 4.1. Cell Culture and Virus

*Spodoptera frugiperda* (Sf9) cells were cultured in Grace’s Insect Medium (Thermo Fisher Scientific, Waltham, MA, USA) and maintained at 27 °C. For large-scale culture, High Five cells (Sino Biological), derived from the ovarian cells of the cabbage looper were propagated in a serum-free medium (Express Five, Thermo Fisher Scientific, Waltham, MA, USA) and maintained at 27 °C. PCV2-LG (GeneBank No.HM038034.1), PCV2-YZ (GeneBank No.EU503040) and PCV2 mAb were preserved by the Laboratory of Microbiology and Immunology, College of Veterinary Medicine, South China Agricultural University. The anti-His antibody was purchased from Beyotime Biotechnology (Shanghai, China), and the codon-optimized gene encoding the TBCap protein of the PCV2 LG strain was synthesized and cloned into the pUC57 plasmid (Sangon Biotech Co., Ltd., Shanghai, China).

### 4.2. Immunofluorescence Assay and Western Blot Analysis

Recombinant baculovirus expressing the fusion protein TBCap was generated according to the instructions for the Bac-to-Bac baculovirus expression system (Thermo Fisher Scientific, Waltham, MA, USA). Sf9 cells were infected with Bac-TBCap at an M.O.I of 2 and then incubated at 27 °C for 72 h. An indirect immunofluorescence assay (IFA) was used to examine the expression of TBCap in Sf9 cells. Sf9 cells were fixed with 4% paraformaldehyde and then detected using an anti-His mouse monoclonal antibody as the primary antibody and FITC-coupled rabbit anti-mouse antibody. In addition, High Five cells were infected with Bac-TBCap at an M.O.I of 2, 5 and 8 to test the level of TBCap expression at different time points after infection. Supernatants and sonicated cell lysates were collected, and the expression of recombinant proteins was detected by SDS-PAGE and Western blotting analysis. Protein bands were detected using the Tanon Fine-do X6 chemiluminescence imaging system.

### 4.3. Assembly and Identification of Virus-like Particles

The above-harvested lysate was purified on a sucrose density gradient (15–60%) at 100,000× *g*, 4 °C for 5 h. The resulting liquid column was fractionated into 2 mL graded fractions and the pellet was resuspended in 1 mL PBS. The obtained VLP fractions were checked by Transmission Electron Microscopy (TEM) via negative staining with 0.5% aqueous uranyl acetate.

### 4.4. Preparation of the Vaccine

Antigens were prepared by infecting High Five cells with Ac-Cap and Ac-TBCap at an M.O.I of 2. At 96 h after infection, the sonicated cell lysates were centrifuged at 11,000× *g* for 15 min at 4 °C. Cap/TBCap was mixed with Montanide ^TM^ Gel 01 ST adjuvant (Seppic, Paris, France) at a 4:1 ratio (*w*/*w*) or with Montanide ISA 201 VG adjuvant (Seppic, Paris, France) at a 1:1 ratio mix (w/o/w) according to the manufacturer’s instructions.

### 4.5. Animal Immunization and Challenge Experiments

For the screening study of the vaccine adjuvant, 30 4-week-old female BALB/c mice (Southern Medical University, Guangzhou, China) were randomly divided into 6 groups of 5 mice each, and vaccines prepared with the two adjuvants as described in 4.4 were administered by subcutaneous multipoint injection, followed by a booster vaccination at the same dose two weeks after the initial immunization (Table 2). Blood was collected at 14, 21 and 28 dpi and sera were isolated and tested for PCV2-specific antibodies by ELISA.

For the pig immunization experiment, 20 piglets tested negative for PCV2 and porcine reproductive and respiratory syndrome virus (PRRSV) antigen and antibody at 1 month of age were randomly divided into 4 groups of 5 piglets each and immunized by intramuscular neck injection; the groups and doses are shown in Table 3. At 3 weeks after the first immunization, the piglets were immunized by the same route and at the same dose, and blood was collected at 14, 21 and 28 dpi and the sera were isolated and tested for specific antibodies by ELISA. At 4 weeks post-immunization, piglets in groups 1 to 3 were injected nasally (1 mL) and intramuscularly (2 mL) with PCV2-YZ (10^5.5^ TCID_50_/mL) and group 4 served as the negative control group. All piglets were injected with 4 mL of keyhole hemocyanin (KLH) emulsion emulsified with Freund’s incomplete adjuvant at multiple points in the 3 d and 6 DPC before the challenge. After the challenge, body temperature was measured at regular times each day and the piglets were observed for clinical performance. Blood was collected at 14 DPC and 28 DPC and sera were isolated for viremia testing. All piglets were dissected at 28 DPC and each pig was observed for organ lesions in organs such as the lungs and lymph nodes. Lymph nodes and lungs were fixed in 4% paraformaldehyde and used for tissue sectioning and immunohistochemistry experiments. The viral load in the lymph nodes was determined by quantitative real-time PCR (qPCR).

For the mouse immunization experiment, 54 4-week-old BALB/c mice were randomly divided into 6 groups of 9 mice each, and the mice were immunized by subcutaneous multi-point injection, and the groups and immunization doses are shown in Table 4. Blood was collected from the tail vein at 0, 14, 21, 28 and 42 dpi, and serum was isolated for specific antibody detection by ELISA. At 2 weeks after 3 immunizations, 4 mice from each group were randomly executed to isolate spleen lymphocytes for the detection of a lymphocyte proliferative response and cytokine release. The remaining mice were attacked with PCV2-YZ strain (10^5.5^ TCID_50_/mL) on the same day, via intraperitoneal injection of 300 µL/each and nasal drip of 50 µL/each. At 0, 7, 14 and 28 DPC, serum and anticoagulated blood were randomly collected from 3 mice in each group for ELISA antibody detection and viremia detection, respectively. The mice were weighed, and the mean daily weight gain was calculated at 0 DPC and 28 DPC, respectively. All mice were dissected and killed at 28 DPC, and their lungs, kidneys and spleens were taken for PCV2 viral load assay.

### 4.6. Lymphocyte Proliferation Assay

The lymphocyte proliferation assay was performed using splenocytes from immunized mice. The splenocytes were resuspended at 5 × 10^6^ cells/mL with the complete medium of RPMI-1640 containing 10% FBS, and seeded into 96-well plates with 100 µL per well. Each sample was stimulated with 100 µL complete medium containing purified PCV2 antigen (20 µg/mL) or 100 µL complete medium alone in triplicate. The proliferative responses were detected by a standard CCK-8 method. The results were expressed as stimulation index (SI), calculated using the following formula: SI = (the mean of OD_570nm_ values of PCV2-stimulated wells)/(the mean of OD_570nm_ values of complete medium-treated wells).

### 4.7. Cytokines Release Assay

On 28 dpi, the lymphocytes were seeded into 96-well plates with 5 × 10^5^ cells per well. Triplicate wells with RPMI-1640 alone and 2 µg purified PCV2 antigen were added for 72 h, and supernatants were collected. The concentrations of IL-4 and IFN-γ were detected by mice IL-4 and IFN-γ ELISA Kits (MultiSciences Bio) following standard procedures. According to the instructions, the assay sensitivities of the IL-4 and IFN-γ ELISA kits were determined to be 0.34 pg/mL and 0.83 pg/mL, with a detection range of 7.81 pg/mL-500 pg/mL and 23.44 pg/mL-1500 pg/mL, respectively. The concentrations of mice IL-4 and IFN-γ in the supernatants were calculated by means of a standard curve generated with serial concentrations of standards.

### 4.8. Pathological and Histopathology Studies

All the pigs were humanely euthanized by injecting an overdose of intravenous sodium pentobarbital and were necropsied on 28 DPC. Samples of superficial inguinal lymph nodes and lungs were collected and fixed in 10% neutral-buffered formalin. Paraffin-embedded tissue samples were stained by HE. Microscopic changes were determined by comparing the tissues of the challenged piglets with those of the control group. 

IHC analysis was performed to determine the presence of PCV2 antigen in lymph nodes. Briefly, the tissue sections were incubated with 1/100 diluted PCV2 mAb at 37 °C for 1 h and then incubated with 1/100 diluted HRP-labeled goat anti-rabbit antibody at 37 °C for 1 h. Freshly prepared diaminobenzidine (Wuhan Servicebio Technology, Ltd., Wuhan, China) was added to the sections at room temperature for color development. Positive cells were quantified using a light microscope (Nikon, Tokyo, Japan).

### 4.9. Statistical Analysis

All statistical analyses were performed using GraphPad Prism Version 8. The data, including antibody response, cytokine production and viral load in tissues, were compared by one-way analysis of variance (ANOVA) among different groups. *p* < 0.05 was considered statistically significant.

## 5. Conclusions

In summary, this study describes the generation, characterization, self-assembly VLP formation and immunogenicity of a recombinant baculovirus incorporating a Cap protein with dominant T and B cell antigenic epitopes. The recombinant protein was able to induce high levels of humoral and cellular immune responses in mice and piglets, partially protected piglets from PCV2 viremia and significantly reduced PCV2 viral load in lymphoid tissues, and the results suggest that the recombinant protein TBCap is expected to be an excellent candidate for PCV2 subunit vaccine development. The introduction of porcine-derived CD154 further enhances the level of biased Th1-type cell-mediated immune response and the CD154+Cap subunit vaccine is an attractive candidate for the prevention and control of PCV2 infection.

## Figures and Tables

**Figure 1 ijms-23-14126-f001:**
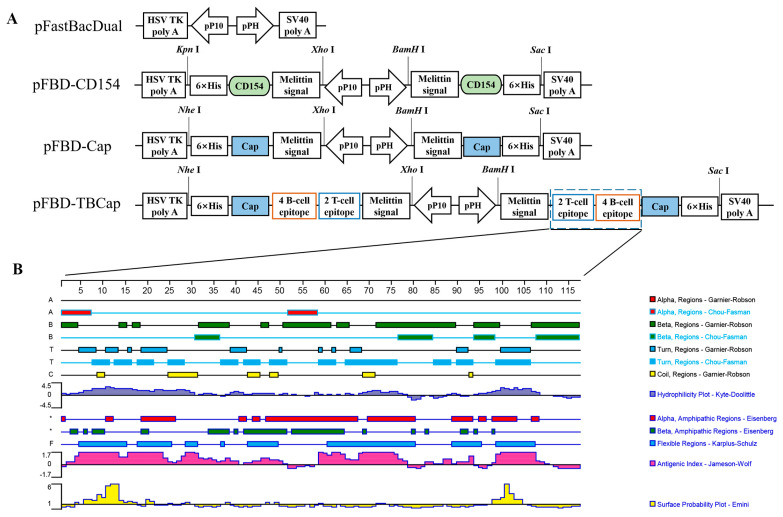
Construction of shuttle plasmids and prediction of antigenic epitope structures. (**A**) Schematic diagram of the shuttle plasmid construction. (**B**) Secondary structure prediction of dominant antigenic epitope peptides.

**Figure 2 ijms-23-14126-f002:**
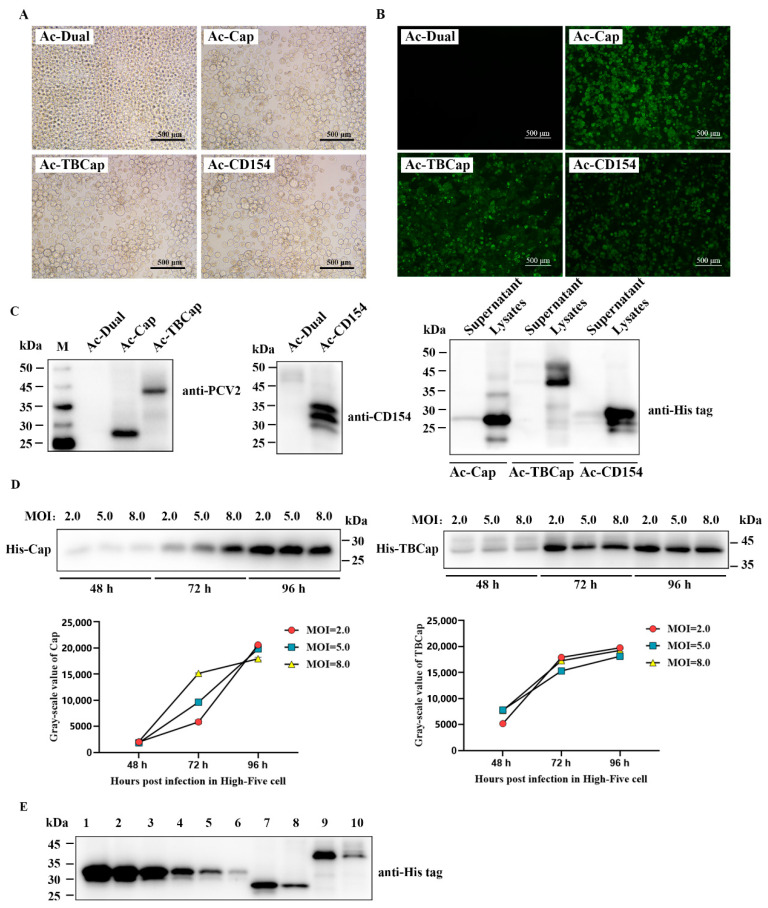
Identification of recombinant protein expression, optimization of conditions and semi-quantitative analysis. (**A**) Normal sf9 cells and sf9 cells showed CPE after transfection. (**B**) Observation of recombinant protein expression by IFA. (**C**) Identification of recombinant protein expression by Western blot. (**D**) Optimization of expression conditions for recombinant proteins. (**E**) Semi-quantitative analysis of recombinant proteins. Lanes 1–6 are, in order, 1000 µg/mL, 750 µg/mL, 500 µg/mL, 250 µg/mL, 100 µg/mL, and 50 µg/mL concentrations of standard protein; lanes 7 and 9 are cell lysates infected with Ac-Cap and Ac-TBCap, respectively; lanes 8 and 10 are cell culture supernatants infected with Ac-Cap and Ac-TBCap, respectively.

**Figure 3 ijms-23-14126-f003:**
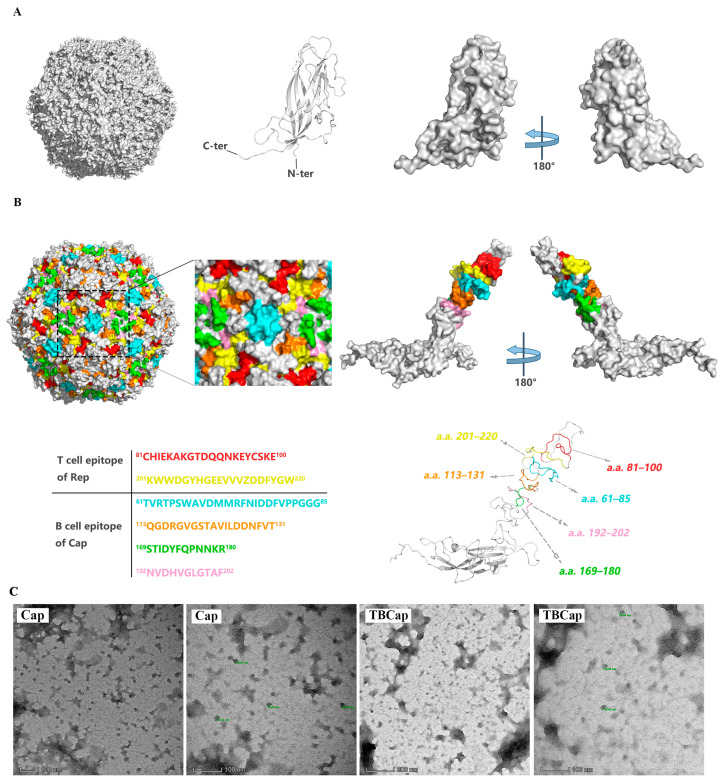
Structural modeling and assembly of PCV2 VLP. (**A**) Three-dimensional reconstruction of the cryo-EM structure of the VLP assembled with the PCV2 cap as the basic unit. Cryo-EM density maps of full-length PCV2 VLPs are shown, with PCV2 VLP density indicated in grey. (**B**) Three-dimensional reconstruction of the cryo-EM structure of the VLP assembled with the TBCap as the basic unit. Again, showing typical icosahedral particle folds, antigenic epitopes on the surface of the PCV2 capsid are shown in different colors and an enlarged region of this capsid is shown. (**C**) The recombinant Cap and TBCap virus-like particles were identified by transmission electron microscopy. VLPs with diameters of approximately 17 nm and 20 nm, respectively, could be observed, consistent with the theoretical size.

**Figure 4 ijms-23-14126-f004:**
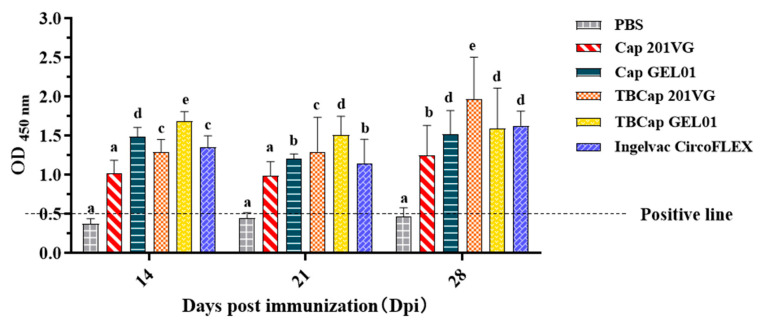
The OD450 of each group at different days post-immunization (dpi) by PCV2 ELISA. Different superscripts (a, b, c, d, e) indicate significant differences among groups (*p* < 0.05).

**Figure 5 ijms-23-14126-f005:**
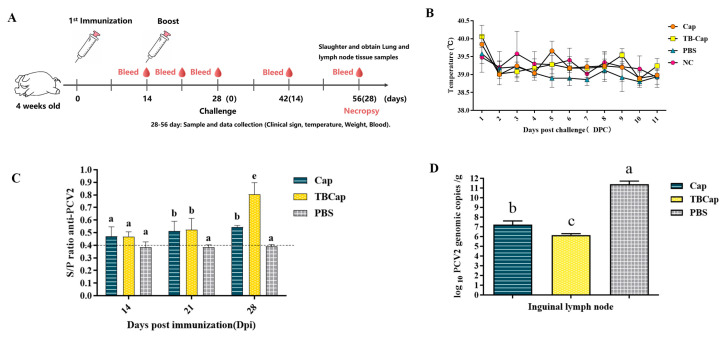
Research on immunogenicity and attack protection tests in piglets. (**A**) Piglet immunization strategy. (**B**) Changes in piglet body temperature after challenge. (**C**) ELISA for PCV2-specific antibodies in piglets after immunization (S/P). (**D**) Lymph node virus load in piglets at autopsy. Different superscripts (a, b, c, e) indicate statistically significant differences in lymph node viral load between the test groups (*p* < 0.05).

**Figure 6 ijms-23-14126-f006:**
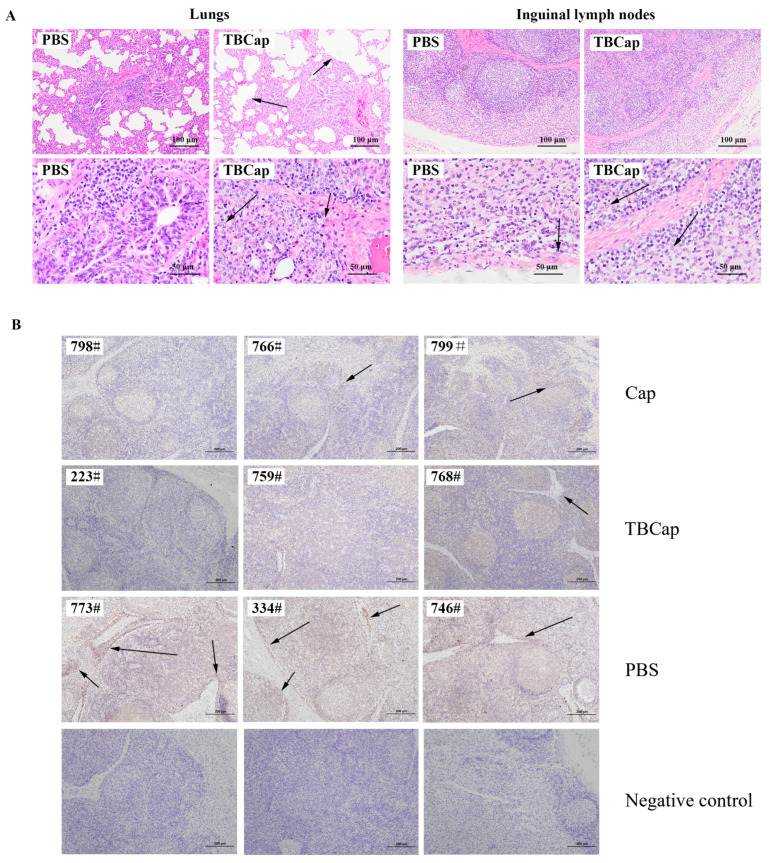
Histopathological testing of piglets after challenge. (**A**) HE staining of piglet lung and inguinal lymph node tissue. (**B**) IHC of inguinal lymph node tissue in piglets, black arrows indicate tan-colored cells in the inguinal lymph node tissue.

**Figure 7 ijms-23-14126-f007:**
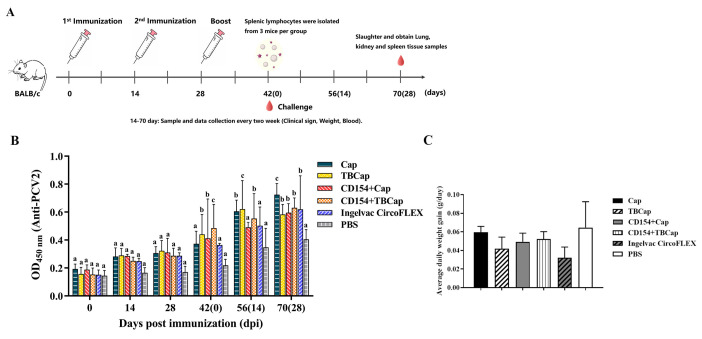
Immunization strategies and immunogenicity in mice. (**A**) Mice immunization strategy. (**B**) ELISA for PCV2-specific antibodies in mice after immunization. (**C**) ADWG of mice after challenge. Different superscripts (a, b, c) indicate statistically significant differences in lymph node viral load between the test groups (*p* < 0.05).

**Figure 8 ijms-23-14126-f008:**
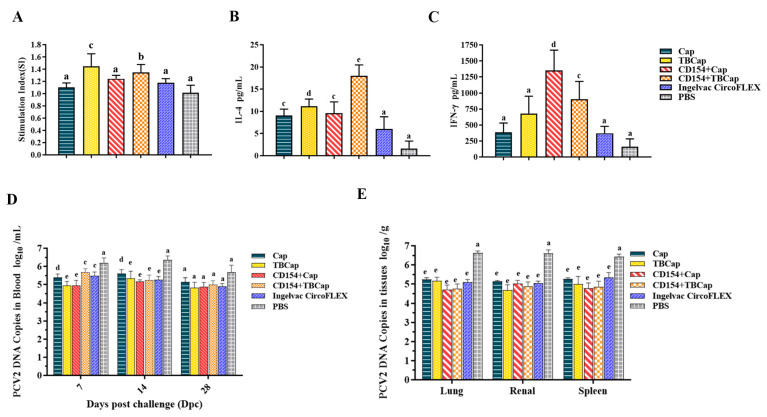
Cytokine secretion and viral load. (**A**) The stimulation of splenic lymphocyte samples collected at 42 dpi. (**B**) Quantitative analysis of interleukin 4 (IL-4) in the supernatant of PCV2-stimulated splenic lymphocytes in immunized mice at 42 dpi. (**C**) Quantification of interferon-gamma (IFN-γ) in the supernatant of PCV2-stimulated splenic lymphocytes in immunized mice at 42 dpi. (**D**) PCV2 DNA loads (copies/mL) in blood collected at 7, 14 and 28 DPC. (**E**) PCV2 DNA loads (copies/g) in tissues collected on the day of necropsy. Different superscripts (a, b, c, d, e) indicate statistically significant differences in lymph node viral load between the test groups (*p* < 0.05).

**Table 1 ijms-23-14126-t001:** Viraemia and clinical signs in piglets after challenge.

Group	No.	Viraemia		Day with Fever (≥40 ℃)	Clinical Symptoms
		14 DPC	28 DPC		
Cap	766	−	−	+	−
	741	−	−	−	+
	798	−	−	−	−
	799	−	−	−	−
	791	−	−	−	−
	Total	0/5	0/5	1/5	1/5
TBCap	780	−	−	+	−
	223	−	−	−	−
	759	−	−	−	−
	768	−	−	−	−
	206	−	−	−	−
	Total	0/5	0/5	1/5	0/5
PBS	790	+	−	−	−
	756	+	−	−	−
	773	+	+	−	++
	746	+	+	−	+++
	334	+	+	−	++
	Total	5/5	3/5	0/5	3/5

“−” indicates negative PCR detection or temperature < 40 °C or no clinical symptoms, “+” indicates positive PCR detection or temperature ≥ 40 °C or clinical symptoms, and the number of “+” indicates the degree.

**Table 2 ijms-23-14126-t002:** Mouse vaccination strategies.

Group	Formulation	Adjuvant	Immunization Time Points (DAI)	Vaccine Dose
1, PBS	PBS	-	0.14	200 μL
2, Cap 201VG	Cap	ISA 201VG	0.14	200 μL
3, Cap Gel 01	Cap	Montanide^TM^ GEL 01	0.14	200 μL
4, TBCap 201VG	TBCap	ISA 201VG	0.14	200 μL
5, TBCap Gel 01	TBCap	Montanide^TM^ GEL 01	0.14	200 μL
6, Ingelvac Circo FLEX	Ingelvac Circo FLEX	-	0.14	200 μL

**Table 3 ijms-23-14126-t003:** The vaccination and challenge status of experimental piglets.

Group	Formulation	Adjuvant	Vaccine Dose	Challenge Dose (TCID_50_)
1, Cap	Cap	Montanide^TM^ GEL 01	1 mL	3 mL, 1 × 10^5.5^
2, TBCap	TBCap	Montanide^TM^ GEL 01	1 mL	3 mL, 1 × 10^5.5^
3, PBS	PBS	-	1 mL	3 mL, 1 × 10^5.5^
4, Negative group	-	-	-	-

**Table 4 ijms-23-14126-t004:** The vaccination and challenge status for mice.

Group	Formulation	Adjuvant	Vaccine Dose	Challenge Dose (TCID_50_)
1, Cap	Cap	Montanide^TM^ GEL 01	200 μL	350 μL, 1 × 10^5.5^
2, TBCap	TBCap	Montanide^TM^ GEL 01	200 μL	350 μL, 1 × 10^5.5^
3, CD154+Cap	CD154, Cap	Montanide^TM^ GEL 01	200 μL	350 μL, 1 × 10^5.5^
4, CD154+TBCap	CD154, TBCap	Montanide^TM^ GEL 01	200 μL	350 μL, 1 × 10^5.5^
5, Ingelvac Circo FLEX	Ingelvac Circo FLEX	-	200 μL	350 μL, 1 × 10^5.5^
6, PBS	PBS	-	200 μL	350 μL, 1 × 10^5.5^

## Data Availability

The data presented in this study are available on request from the corresponding author. The data are not publicly available due to intellectual property considerations.

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
