# Peer review of "Immunogenicity and Immunoprotection of PCV2 Virus-like Particles Incorporating Dominant T and B Cell Antigenic Epitopes Paired with CD154 Molecules in Piglets and Mice"

_ijms, 2022, doi:10.3390/ijms232214126_

Round 1

Reviewer 1 Report

In the paper the authors expressed and characterized full-length PCV2 Cap proteins fused to dominant T and B cell antigenic epitopes and porcine-derived CD154 molecules using baculovirus. They found that the fused epitopes of Cap proteins were capable of forming virus-like particles (VLPs). The recombinant protein was able to induce high levels of humoral and cellular immune responses in mice and piglets, reduced PCV2 viremia and viral load in lymphoid tissues. The introduction of porcine-derived CD154 further enhances the level of biased Th1-type cell-mediated immune response

Comments:

 Information is missing on whether the authors have obtained ethical approval from an ethics committee to carry out study with the use of animal  

Animal experiments – clinical score assessment is missing

Line 18 : …..the global pig industry. the nucleocapsid protein ….. start a sentence with a capital letter , it should be corrected

Line 24 : ….. virus-like particles (VLPs). higher levels of humoral  ….. start a sentence with a capital letter , it should be corrected virus-like particles (VLPs).

Lines 39-41: PCV2 is arguably the most economically important pathogen affecting the global pig industry - I find this wording exaggerated, it is an important pathogen but hardly the most important, besides there are currently vaccines that are quite effective in protecting against the development of PCVAD.

Line 202-203: Body temperatures of piglets were measured for Two (it should be two) consecutive weeks after the attack – inoculation?

Table 1. Viraemia and clinical signs in piglets after challenge – in the M&M section there is no explanation for the group entry 4?

Line 230 - start a sentence with a capital letter

Line 338-341 – this fragment should be deleted

Line 494: Titers of IFN-γ and … are you talking about concentration?

4.7 Cytokines release assay – the characteristics of the ELISA assay is lacking e.g. detection limit.

Acknowledgments: In this section, you can acknowledge any support given which is not covered 534 by the author contribution or funding sections. This may include administrative and technical sup-535 port, or donations in kind (e.g., materials used for experiments – this should be deleted

Author Response

Dear Editors and Reviewers:

Thank you for your letter and for the reviewers’ comments concerning our manuscript entitled “Immunogenicity and Immunoprotection of PCV2 Virus-like Particles Incorporating Dominant T and B Cell Antigenic Epitopes Paired with CD154 Molecules in Piglets and Mice” (ID: ijms-1996188). Those comments are all valuable and very helpful for revising and improving our paper, as well as the important guiding significance to our researches. We have studied comments carefully and have made correction which we hope meet with approval. Revised portion are marked in red with track change mode in the paper. The main corrections in the paper and the responds to the reviewer’s comments are as flowing:

Point 1: Information is missing on whether the authors have obtained ethical approval from an ethics committee to carry out study with the use of animal

Response 1: We are very sorry for our negligence of the information regarding Ethics Committee or Institutional Review Board approval, and we have supplemented relevant information as required in the part of “Ethical Statement” with track change mode in the revised manuscript.

Point 2: Animal experiments – clinical score assessment is missing

Response 2: Thank you very much for your helpful advice, we also consider whether clinical score is needed, we reviewed a lot of literature and combined with pre laboratory studies, the main clinical symptoms of piglets after PCV2 infection are coat coarsening, appetite loss, etc., but this indicator alone is not enough to fully represent the degree of PCV2 infection and has a certain subjective judgment. Therefore, as shown in Table 1 and Figure 6, we used "-" for no clinical symptoms, "+" for clinical symptoms, and the number of "+" for the degree of clinical symptoms. Furthermore, we comprehensively analyzed the viremia detected by PCR, body temperature of the piglets after infection, lung and lymph node histopathology observed by HE staining, and it is worth mentioning that the IHC findings of lymph node tissues in Figure 6B, namely, the brown yellow brown staining of macrophages and some lymphocytes in the lymph nodes containing the PCV2 antigen, which are the gold standard for the detection of PCV2 infection in piglets. In addition, as it is generally accepted that mice infected with PCV2 do not develop overt clinical symptoms, we evaluated the protective efficacy of mice immunized with PCV2 infection by measuring PCV2 viral loads in blood, lungs, kidneys, and spleen.

Point 3:

Line 18 : …..the global pig industry. the nucleocapsid protein ….. start a sentence with a capital letter , it should be corrected

Line 24 : ….. virus-like particles (VLPs). higher levels of humoral  ….. start a sentence with a capital letter , it should be corrected virus-like particles (VLPs).

Line 230 - start a sentence with a capital letter

Line 338-341 – this fragment should be deleted

Response 3: We are very sorry for our negligence and the nonstandard writing, and we have tried our best to improve the manuscript and made a thorough revision in the manuscript. These changes will not influence the content and framework of the paper, and here we have not listed all of the changes but marked in red in revised manuscript.

Point 4: Lines 39-41: PCV2 is arguably the most economically important pathogen affecting the global pig industry - I find this wording exaggerated, it is an important pathogen but hardly the most important, besides there are currently vaccines that are quite effective in protecting against the development of PCVAD.

Response 4: We are very sorry for our inaccurate description. Considering the Reviewer’s suggestion, we have changed the previously inappropriate description to“PCV2 is arguably an important pathogen affecting the global pig industry and the main pathogen of PCVAD” with track change mode in Lines 39-41.

Point 5: Line 202-203: Body temperatures of piglets were measured for Two (it should be two) consecutive weeks after the attack – inoculation?

Response 5: We are very sorry for our inaccurate description, and what we were going to say was “Body temperatures of piglets were measured for two consecutive weeks after the challenge”, and we have changed it with track change mode in the revised manuscript.

Point 6: Table 1. Viraemia and clinical signs in piglets after challenge – in the M&M section there is no explanation for the group entry 4?

Response 6: We are very sorry for our negligence, “the group entry 4” should be the group of PBS, and we have change it in the table 1.

Point 7: Line 494: Titers of IFN-γ and … are you talking about concentration?

Response 7: We are very sorry for the inaccurate description, and what we had originally intended to express was indeed the concentration of IFN-γ and IL-4, in line 511, we have changed “Titers” in “Concentration” with track change mode in the revised manuscript.

Point 8: 4.7 Cytokines release assay – the characteristics of the ELISA assay is lacking e.g. detection limit.

Response 8: We are very sorry for our unclear description. Actually, all groups contained a certain content of IFN- γ and IL-4, the aim of this experiment was to calculate the actual content of IFN- γ and IL-4 in the supernatant using a standard curve generated with serial concentration standards provided in the ELISA Kit, and compared with the PBS group or the commercial vaccine group by statistical analysis. Therefore, we did not use the detection line to adjudicate the negative and positive of cytokines, and in line 513-514, we have further enriched the description of these methods with track change mode in the revised manuscript.

Point 9: Acknowledgments: In this section, you can acknowledge any support given which is not covered 534 by the author contribution or funding sections. This may include administrative and technical sup-535 port, or donations in kind (e.g., materials used for experiments – this should be deleted

Response 9: We are very sorry for our negligence, and we have deleted the part of Acknowledgments with track change mode in the revised manuscript (in blue).

We appreciate for your warm work earnestly, and hope that the correction will meet with approval.

Once again, thank you very much for your comments and suggestions.

Author Response

Dear Editors and Reviewers:

Thank you for your letter and for the reviewers’ comments concerning our manuscript entitled “Immunogenicity and Immunoprotection of PCV2 Virus-like Particles Incorporating Dominant T and B Cell Antigenic Epitopes Paired with CD154 Molecules in Piglets and Mice” (ID: ijms-1996188). Those comments are all valuable and very helpful for revising and improving our paper, as well as the important guiding significance to our researches. We have studied comments carefully and have made correction which we hope meet with approval. Revised portion are marked in red with track change mode in the paper. The main corrections in the paper and the responds to the reviewer’s comments are as flowing:

Point 1: Lots of typographical errors are observed throughout the manuscript. For example: In introduction, the line number 33 needs to corrected

Point 2: The binomial scientific names should always be italicized. The authors should correct the following line numbers

           (i). 52

           (ii). 70

Response: We are very sorry for our negligence and the nonstandard writing, and we have tried our best to improve the manuscript and made a thorough revision in the manuscript. These changes will not influence the content and framework of the paper, and here we have not listed all of the changes but marked in red in revised paper.

Point 3: Is the vaccine formulations are showing any toxic adverse effects and what is the safety profile of formulations?

Point 4: What is the stability and storage conditions of vaccine formulations?

Response: In fact, we have done these two experiments. However, due to the length of the article, this part has been omitted. Considering the Reviewer’s suggestion, in line 182-198, we have supplemented the description of the results of the vaccine safety and stability experiments in the vaccine quality detection part with track change mode in the revised manuscript.

We appreciate for your warm work earnestly, and hope that the correction will meet with approval.

Once again, thank you very much for your comments and suggestions.

Round 2

Reviewer 1 Report

The authors have correctly addressed my comments and improved the manuscript.  However, the detection limit of ELISAs used to analyse cytokine concentrations should be stated in the article.  

Author Response

Dear Editors and Reviewers:

Thank you very much for your helpful advice, considering the Reviewer’s suggestion, we have added descriptions about cytokine assay sensitivity and detection range in lines 508-510. In addition, to be consistent with the order in which the results were described, we switched the order of appearance of “IFN-γ” and “IL-4” in the methods in lines 506-511. Revised portion are marked in red with track change mode in the paper.

We appreciate for your warm work earnestly, and hope that the correction will meet with approval.